# Long-term alterations in brain and behavior after postnatal Zika virus infection in infant macaques

Jessica Raper [1,2], Zsofia Kovacs-Balint[1], Maud Mavigner[2,3], Sanjeev Gumber[1], Mark W. Burke[4], Jakob Habib[2], Cameron Mattingly[2], Damien Fair [5], Eric Earl [5], Eric Feczko[5], Martin Styner [6], Sherrie M. Jean[1], Joyce K. Cohen[1,7], Mehul S. Suthar[1,3,8], Mar M. Sanchez[1,7], Maria C. Alvarado [1] & Ann Chahroudi [1,2,3✉]

Zika virus (ZIKV) infection has a profound impact on the fetal nervous system. The postnatal period is also a time of rapid brain growth, and it is important to understand the potential neurobehavioral consequences of ZIKV infection during infancy. Here we show that postnatal ZIKV infection in a rhesus macaque model resulted in long-term behavioral, motor, and cognitive changes, including increased emotional reactivity, decreased social contact, loss of balance, and deficits in visual recognition memory at one year of age. Structural and functional MRI showed that ZIKV-infected infant rhesus macaques had persistent enlargement of lateral ventricles, smaller volumes and altered functional connectivity between brain areas important for socioemotional behavior, cognitive, and motor function (e.g. amygdala, hippocampus, cerebellum). Neuropathological changes corresponded with neuroimaging results and were consistent with the behavioral and memory deficits. Overall, this study demonstrates that postnatal ZIKV infection in this model may have long-lasting neurodevelopmental consequences.

[1] Yerkes National Primate Research Center, Emory University, Atlanta, GA, USA. [2] Department of Pediatrics, Emory University School of Medicine, Atlanta, GA, USA. [3] Center for Childhood Infections and Vaccines of Children's Healthcare of Atlanta and Emory University, Atlanta, GA, USA. [4] Department of Physiology and Biophysics, Howard University, Washington, DC, USA. [5] Oregon Health and Science University, Portland, OR, USA. [6] Department of Psychiatry, University of North Carolina, Chapel Hill, NC, USA. [7] Psychiatry & Behavioral Sciences, Emory University School of Medicine, Atlanta, GA, USA. [8] Emory Vaccine Center, Atlanta, GA 30329, USA. ✉email: achahro@emory.edu

Zika virus (ZIKV) is a neurotropic flavivirus that was first isolated in 1947 from rhesus macaques (RMs) in Uganda[1] and then from humans in 1952[2]. As of May 2019, ZIKV had infected individuals in 89 countries and territories, including the United States[3]. ZIKV is transmitted primarily by the bite of an *Aedes* genus mosquito, but also through sexual contact, blood transfusion, organ transplantation, and from mother to fetus during gestation. While the majority of ZIKV infections in adults are either asymptomatic or result in mild symptoms, ZIKV is associated with severe birth defects when women are infected during pregnancy[4–6]. The passage of ZIKV into the brain and its ability to infect human cortical neural progenitor cells[7] induces neuropathological changes that have been reported since the late 1950s[8,9]. However, it was not until the 2015–2016 outbreak of ZIKV in Brazil that the link to microcephaly was first established[5], leading to the declaration of a global health emergency[10]. Congenital infection with ZIKV occurs throughout gestation with resultant microcephaly and other brain malformations[11–14] that are thought to be the consequence of ZIKV infection of neural progenitor cells as well as activation of innate immune responses[7,15]. Congenital ZIKV syndrome is a pattern of birth defects that includes severe microcephaly, thinning of the cerebral cortex with subcortical calcifications, macular scarring and retinal mottling, congenital contractures, and hypertonicity[6]. Infants born with this syndrome can develop seizures, hearing and vision problems, feeding difficulties, and gross motor abnormalities[16]. Several reports have documented the postnatal onset of microcephaly, neurologic dysfunction, and neurodevelopmental abnormalities in infants infected prenatally but without overt symptoms at birth[17–21]. A recent follow-up of 216 toddlers with prenatal ZIKV exposure reported microcephaly in only 3.7% cases, yet despite no signs of congenital ZIKV syndrome 40.4% had below average scores on the Bayley-III neurodevelopmental evaluation[22]. Together these studies highlight the potential of ZIKV to cause ongoing damage in the postnatal period.

Infants and children can also acquire ZIKV infection postnatally through mosquito bites and possibly breast milk[23]. As ZIKV has adapted to persistent endemic transmission[24], continued exposures of the young are likely, evidenced by 9% of children aged 1–4 years testing ZIKV seropositive in Indonesia[25]. There have been conflicting reports on the neurologic complications and neurodevelopmental outcomes of children exposed to ZIKV in the peripartum and postnatal periods[26–30]. Studies with large numbers of cases have yet to be published, while research in animal models suggest that ZIKV infection during the postnatal period may also have a deleterious impact on development[31,32]. We recently demonstrated in infant RMs that postnatal infection with ZIKV disseminates to the central and peripheral nervous systems with histologic features in the post-mortem brain similar to prenatal ZIKV infection[31]. Despite clearance of virus from blood after 7 days, brain structural and functional anomalies were detected up to 6 months after postnatal infection, including ventriculomegaly, blunted hippocampal growth, and weaker amygdala–hippocampus functional connectivity (FC) compared to uninfected infants. Further, the altered FC between these brain limbic structures was associated with atypical behavioral responses. Yet, little is known about the longer-term consequences of postnatal ZIKV infection on brain and behavior. Given the quick clearance of ZIKV in the periphery and the considerable protracted development of the primate brain, there is considerable opportunity for plasticity and compensation of function.

The current study examines the impact of early postnatal ZIKV infection on brain and behavior up to 12 months of age (equivalent to 4–5 years in human age), permitting a longitudinal comparison to results previously obtained at 3 and 6 months of age. We demonstrate that postnatal ZIKV infection results in long-term behavioral differences, abnormal brain structure and function, and corresponding brain pathological changes, therefore suggesting a lack of compensatory plasticity or functional recovery.

## Results

**Persistent changes in socioemotional behaviors**. Two infant RMs (ZIKV-5 and ZIKV-6) were challenged subcutaneously with $10^5$ plaque-forming units (pfu) of ZIKV PRVABC59 at 5 weeks of age (equivalent to a 4-month-old human infant[33]) and two infant RMs served as age-, sex-, and rearing-matched uninfected controls (Control-1 and Control-2). To assess the long-term impact of postnatal ZIKV infection on brain development and socioemotional behavior, at 12 months of age, we placed the familiar pairs of RM infants in a novel social play cage (390 cubic feet) on six separate testing days for 4.5 h of total observations of social interactions. Both groups habituated to the novel environment over time, but ZIKV-infected RMs took longer to enter the novel play cage (Fig. 1a), primarily stayed on the perimeter of the cage (Fig. 1b), and rarely explored the center of the cage (Fig. 1c) as compared to controls. The large play cage provided the RM infants room to run, jump, swing, and play, as well as created opportunities for infants to exhibit uncoordinated movements resulting in losses of balance. Both groups spent similar amounts of time playing (Supplementary Table 1), but as they became more comfortable in this large novel environment both groups exhibited an increased number of losses of balance or missteps when jumping and landing long distances (5–8 feet) in the play cage. ZIKV-infected RM infants tended to exhibit more losses of balance compared to controls (Fig. 1d). On the first day, both pairs spent most of their time near each other (Fig. 1e), but by the second day ZIKV-infected infants began spending most of their time away from their familiar partner (Fig. 1f; average across all days $M = 34.8$ and $M = 10.98$, for ZIKV and controls respectively). This decreased prosocial behavior was also evident by ZIKV-infected infants exhibiting less affiliative behaviors as compared to controls (Fig. 1g). Interestingly, ZIKV-5 exhibited increased hostility across testing days (Fig. 1h), which resulted in increased submissive behaviors from ZIKV-6 by the last day of testing (Fig. 1i). Controls and ZIKV-infected RMs exhibited similar levels of anxiety, vocalizations, and stereotypies (Supplementary Table 1). These results show that ZIKV infection at 5 weeks of age resulted in atypical socioemotional behavior and impaired motor function at 12 months of age (equivalent to 4–5 years in humans). Taken together these findings suggest that ZIKV infection during infancy resulted in quantifiable behavioral alterations at 12 months of age that may be consequences of the brain growth and function effects reported below.

**Deficits in memory function**. Next, we examined the potential impact on cognition using a visual recognition memory task. RM infants were trained to sit in a primate chair and viewed color images for the visual paired comparison (VPC) task for object and visuospatial memory, and their looking patterns were recorded using a Tobii T60/T120 eye tracker (Tobii Technology). The VPC task (Fig. 2a) presents a sample image for an accumulative 15 s of "looking time", followed by a delay ranging from 10 to 120 s. In the retention test, the "familiar" sample and a novel image are presented side-by-side for 5 s, and then re-presented with the left/right positions of the images switched. The eye tracker calculated total duration of fixation times to the familiar or novel stimulus, which was used to calculate a novelty preference score [looking time(novel)/total looking time (novel + familiar) × 100], as macaques, like humans, have a natural preference for novelty.

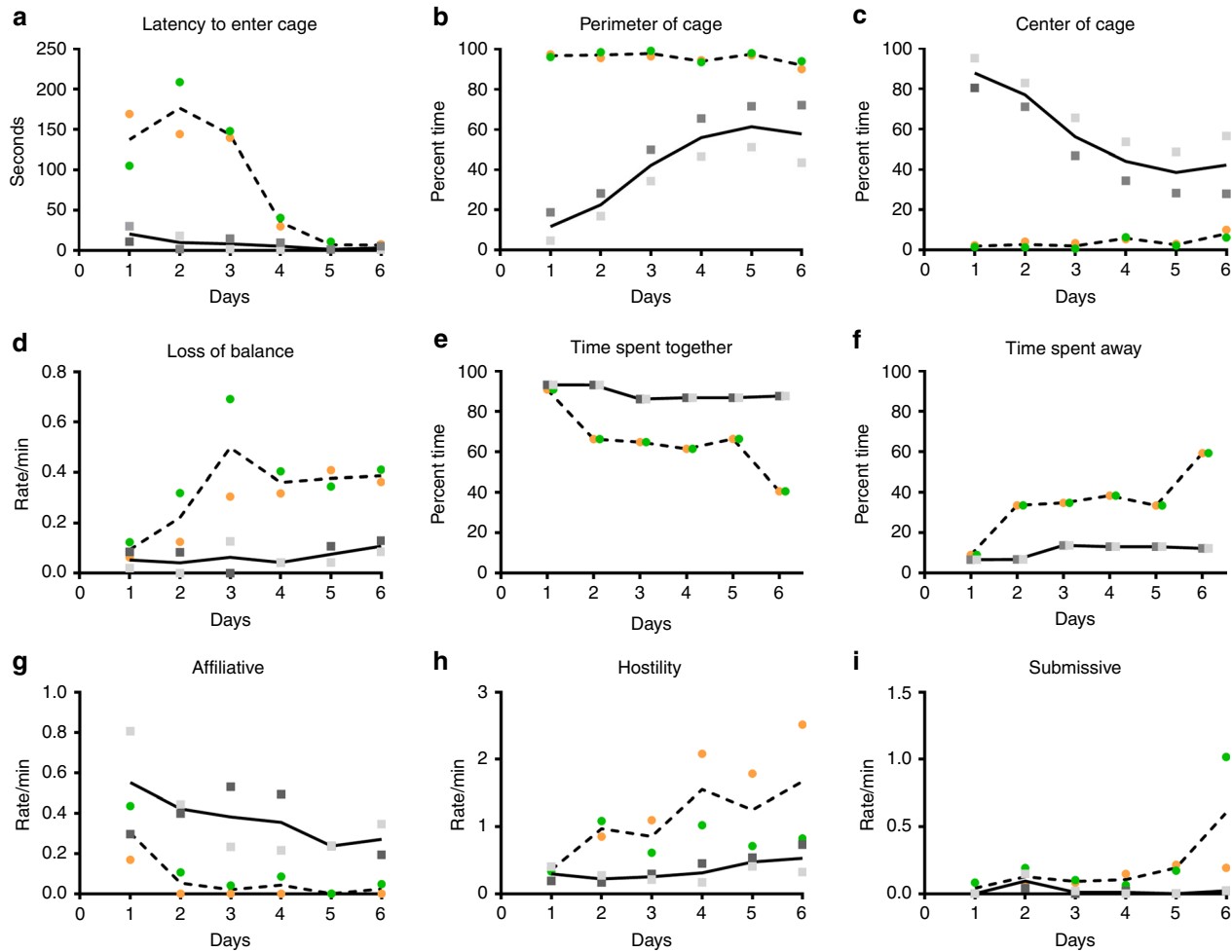

**Fig. 1 Altered socioemotional behavior in infant RMs with postnatal ZIKV infection. a–i** Illustrates the change in behavioral responses of ZIKV-infected infant RMs (dashed line with circles; green = ZIKV-5 and orange = ZIKV-6) and controls (solid line with squares; dark gray = Control-1 and light gray = Control-2) across 6 days of testing in a large novel play cage. Latency to enter the cage (**a**) was measured in seconds, while specific behaviors were measured as rates per min (**d**, **g–i**) or as a percentage of total observation time (**b**, **c**, **e**, **f**).

As shown in Fig. 2b, the RM infants showed similar novelty preference across the 4 delays. However, control infants' performance differed reliably from chance even at the 120 s delay, whereas the ZIKV-infected infants showed a drop in performance at the 120 s delay, such that their performance did not differ from chance at that delay, suggesting a problem with memory retention in this object recognition task.

Although object recognition can be demonstrated early in life, spatial relational memory matures between 12 and 18 months in RM[34]. That is, the ability to learn not only the identities of a set of co-presented items, but also to learn and remember their spatial arrangement. We assessed spatial relational memory using the Object in Place versus Object Control variants of the VPC task (Fig. 2a). For this task, an array of five objects is presented during the Sample phase as for standard VPC. The array disappears for a brief 5 s delay, and then reappears paired with the same array with either three objects repositioned (Object in Place), or three of the objects replaced with novel ones (Object Control). In the present study, following completion of the Object Recognition VPC task, the RM infants were presented with mixed trials of Object in Place and Object Control. As is expected for this age, all four RM infants performed at chance on the Object in Place trials. Although the two control infants had a higher novelty preference on Object Control trials, ZIKV-infected infants remained at chance (Fig. 2c). Given that ZIKV infants exhibited normal

performance at short delays on the standard VPC task, this latter finding suggests a specific deficit in visuospatial processing. This result could reflect that ZIKV-infected infants are unable to perform with an increasing memory load, or it is possible that ZIKV-infected infants may not have adequately sampled all items resulting in lower performance. Overall, the impaired cognitive function detected on the VPC task suggest that postnatal ZIKV infection may have negatively impacted the development of brain areas important for visual learning and memory.

**Persistent structural changes in specific brain regions**. Given the alterations detected in socioemotional behavior and cognitive functioning at 12 months of age, we then conducted magnetic resonance imaging (MRI) to determine potential structural alterations caused by postnatal ZIKV infection in brain regions that control these functions. MRI findings were considered descriptive in nature due to the small sample size; however, a consistent effect was considered potentially clinically relevant. T1- and T2-weighted structural MRI scans revealed no differences between ZIKV-infected RMs and controls in total brain volume (TBV) or intracranial volume (ICV; Supplementary Table 2). Despite similar TBV, the lateral ventricle volumes were larger in ZIKV-infected infant RMs than controls (Fig. 3a–c, Supplementary Fig. 1). In contrast to MRI findings we reported in these RM

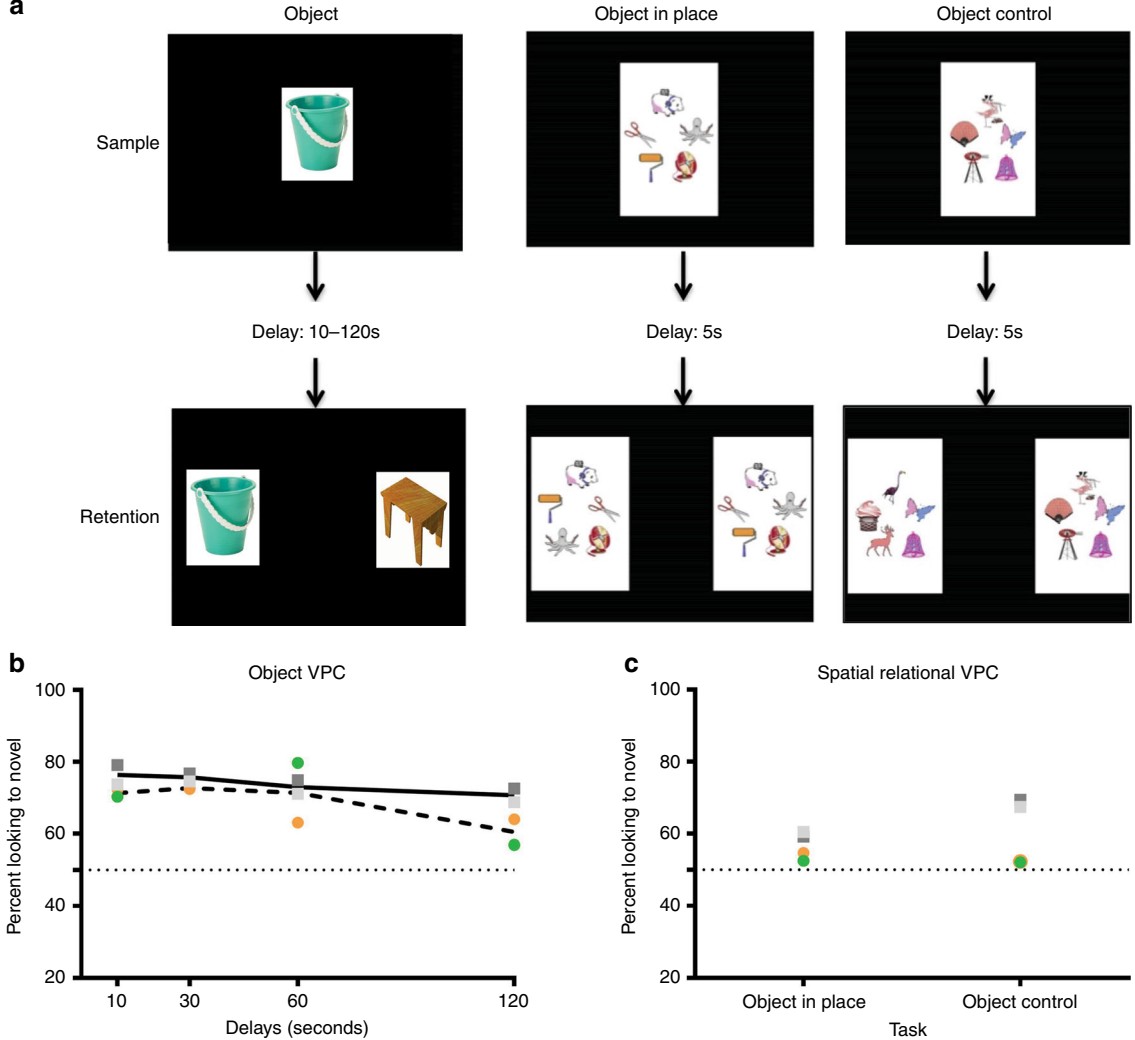

**Fig. 2 Impaired visual recognition memory after postnatal ZIKV infection of infant RMs. a** Illustrates an example trial for each of the three visual paired comparison (VPC) tasks used in this study. **b** Illustrates the percent looking toward the novel stimulus across delays for ZIKV-infected infant RMs (dashed line with circles; green = ZIKV-5 and orange = ZIKV-6) and controls (solid line with squares; dark gray = Control-1 and light gray = Control-2). **c** Compares the percent looking to novelty during object in place versus object control trials. Chance performance is depicted by a dotted line at 50%.

infants at 3 and 6 months of age[31], groups did not differ in GM or WM volumes at 12 months of age, which implies potential neural recovery after postnatal ZIKV insult.

Temporal lobe structures, such as the amygdala and hippocampus, play an important role in socioemotional behavior. They also exhibit a prolonged postnatal development which can make them particularly vulnerable to early central nervous system (CNS) insults, including ZIKV infection. Amygdala volume in RMs increases rapidly from birth to 8 months of age, but continues growing into adulthood[35,36]. Amygdala volumes were similar to controls at 3 and 6 months of age[31]; however, ZIKV-infected RM infants exhibited smaller amygdala volume than controls at 12 months of age (Fig. 3d). Postnatal development of the hippocampus is more protracted than the amygdala, with the hippocampus doubling its volume between 1 week and 2 years of life in RMs[35]. Here, ZIKV-infected RM infants also exhibited smaller hippocampal volumes compared to controls at 12 months of life (Fig. 3e). Decreased hippocampal volumes were previously detected at 3 and 6 months of age[31], and the magnitude of the observed volumetric difference appears to lessen with age, which again may suggest a potential for recovery or compensation during this period of neural plasticity and protracted brain development.

The dorsal striatum (e.g. caudate and putamen) receives inputs from amygdala, hippocampus, prefrontal cortex (PFC), ventral tegmental area, and substantia nigra, making it an important interface for the motor control, reward, and addiction circuity[37]. The striatum has also been implicated in learning and memory and inhibitory control of action[38]. Interestingly, ZIKV-infected RM infants showed smaller putamen volumes compared to controls (Fig. 3f), but no difference in the caudate measurements (Supplementary Table 2). Coordination of motor function is also controlled by the cerebellum, which has been shown to exhibit hypoplasia in cases of congenital Zika syndrome[39]. Our results are consistent with those reports, since at 12 months of age, ZIKV-infected RM infants had decreased cerebellar GM volume than controls (Fig. 3g), although WM volume was unaffected (Supplementary Table 2).

While visual areas are well developed at birth in RMs, synaptic density in visual cortex increases in the first few months of life, followed by a sharp pruning before the first year[40,41]. Visual areas undergo experience-based refinements throughout the first few years of age, indicating selective survival of certain synapses[41]. ZIKV-infected infants had larger GM volumes in the right occipital lobe (Supplementary Table 2), suggesting that they may

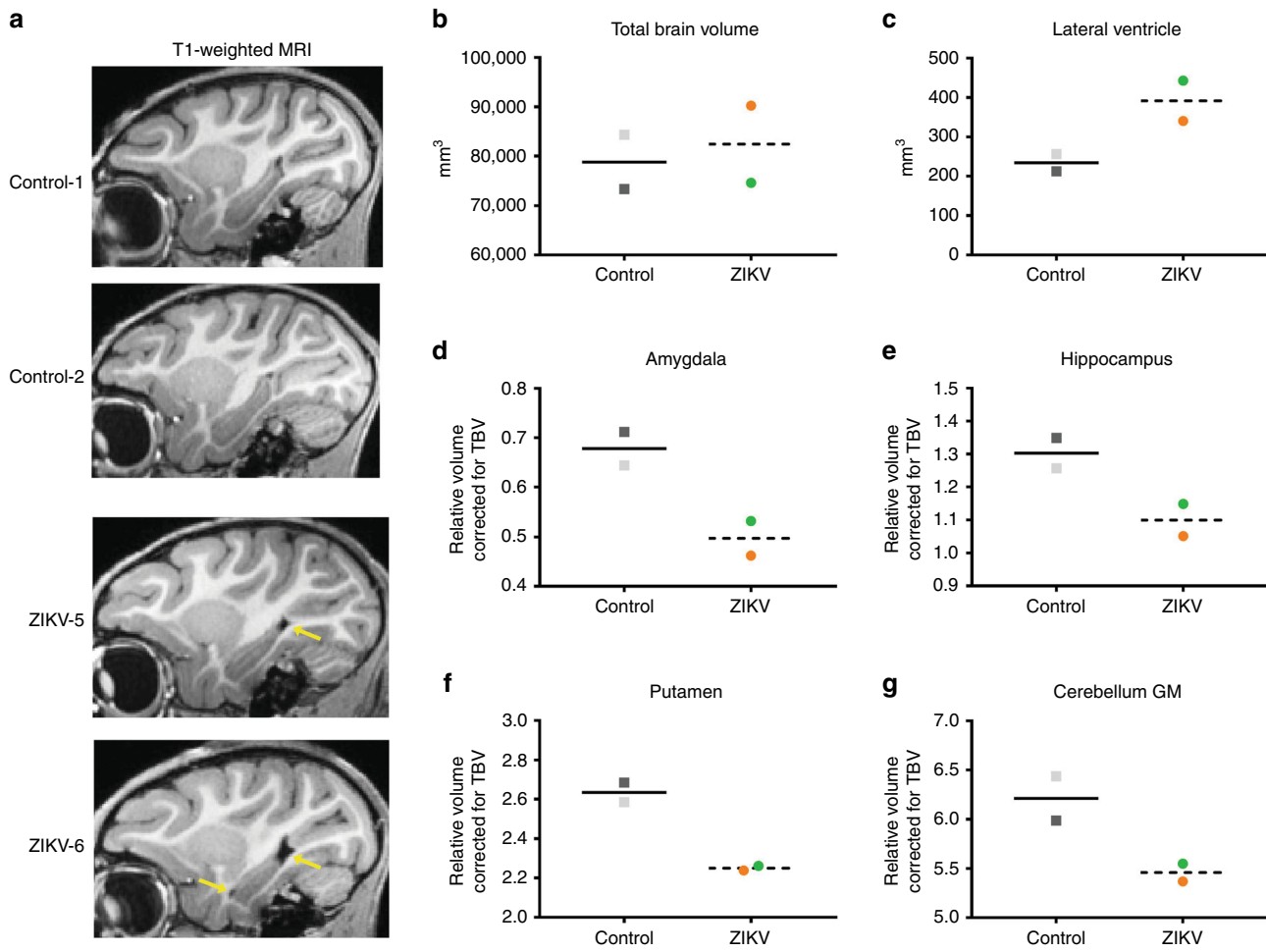

**Fig. 3 Postnatal ZIKV infection leads to ventricle enlargement and reduced subcortical volumes. a** Sagittal T1-weighted structural MRI images through the hippocampus; yellow arrows illustrate the increased lateral ventricle volume in ZIKV-infected RMs. Volumetric measurements of **b** the total brain volume and **c** lateral ventricles in cubic millimeters, as well as **d** amygdala, **e** hippocampus, **f** putamen, and **g** cerebellum gray matter (GM) corrected by total brain volume. The correction used was specific brain region/TBV × 100. ZIKV-infected infant RMs are represented by dashed line with circles (green = ZIKV-5 and orange = ZIKV-6) and controls are represented by solid line with squares (dark gray = Control-1 and light gray = Control-2).

be experiencing less pruning/refinement in visual areas compared to uninfected normally developing controls.

In summary, these structural MRI results indicate that ZIKV infection during infancy can lead to alterations in brain development and growth that are detected as early as 3 months of age[31] and many of which persist through 12 months of age, with other regions showing evidence of recovery.

**Changes in resting state FC.** To assess the long-term impact of postnatal ZIKV infection on FC between brain regions, we conducted resting state functional MRI (rs-fMRI) scans at 12 months of age. Regions of interests (ROIs) were defined using published anatomical parcellations[42] mapped onto the University of North Carolina (UNC)–Emory RM infant atlases registered to the F99 space[43]. First, we examined whether the regions showing structural changes above exhibit parallel alterations in FC, as measured by resting state blood oxygen level-dependent (BOLD) signal-correlated fluctuations (Supplementary Table 3). Although both groups showed positive FC between the amygdala and hippocampus, ZIKV-infected RM infants had weaker FC compared to uninfected controls in the left hemisphere (Fig. 4a, b), which was first detected at 6 months of age[31]. Regions of the PFC, including the orbitofrontal cortex (OFC), dorsolateral PFC (dlPFC), and medial PFC (mPFC) play important roles in

attention, emotional regulation, as well as learning and memory function[44,45]. Controls exhibited stronger FC between right hippocampus and ventromedial OFC area 14 than ZIKV-infected RMs (with one of the infants exhibiting an anticorrelation—negative FC between these regions; Fig. 4c). Other regions where ZIKV animals showed weaker FC than controls include right hippocampus with dlPFC area 46 (Fig. 4d) and between inferior temporal cortex area (TEO) and putamen (Fig. 4e). In contrast, stronger FC was found in ZIKV-infected than control infants between the right amygdala and dlPFC area 46 (Fig. 4f) and between right amygdala and caudate (Fig. 4g). These rs-fMRI data suggest that postnatal ZIKV infection leads to long-term impact in FC between cortico-limbic regions, including PFC, amygdala, hippocampus, and striatal areas, which may affect the functions they control (i.e. emotional regulation, social motivation, motor control, and cognitive maturation).

**Neurohistopathological alterations after postnatal ZIKV.** Once behavioral and neuroimaging assessments were completed the four infant RMs were euthanized to assess neurohistopathology approximately one year after early postnatal ZIKV infection. We have previously reported that ZIKV RNA was cleared from plasma in the two infected RM infants within 7 days post-infection and, while ZIKV RNA was found in the brain at

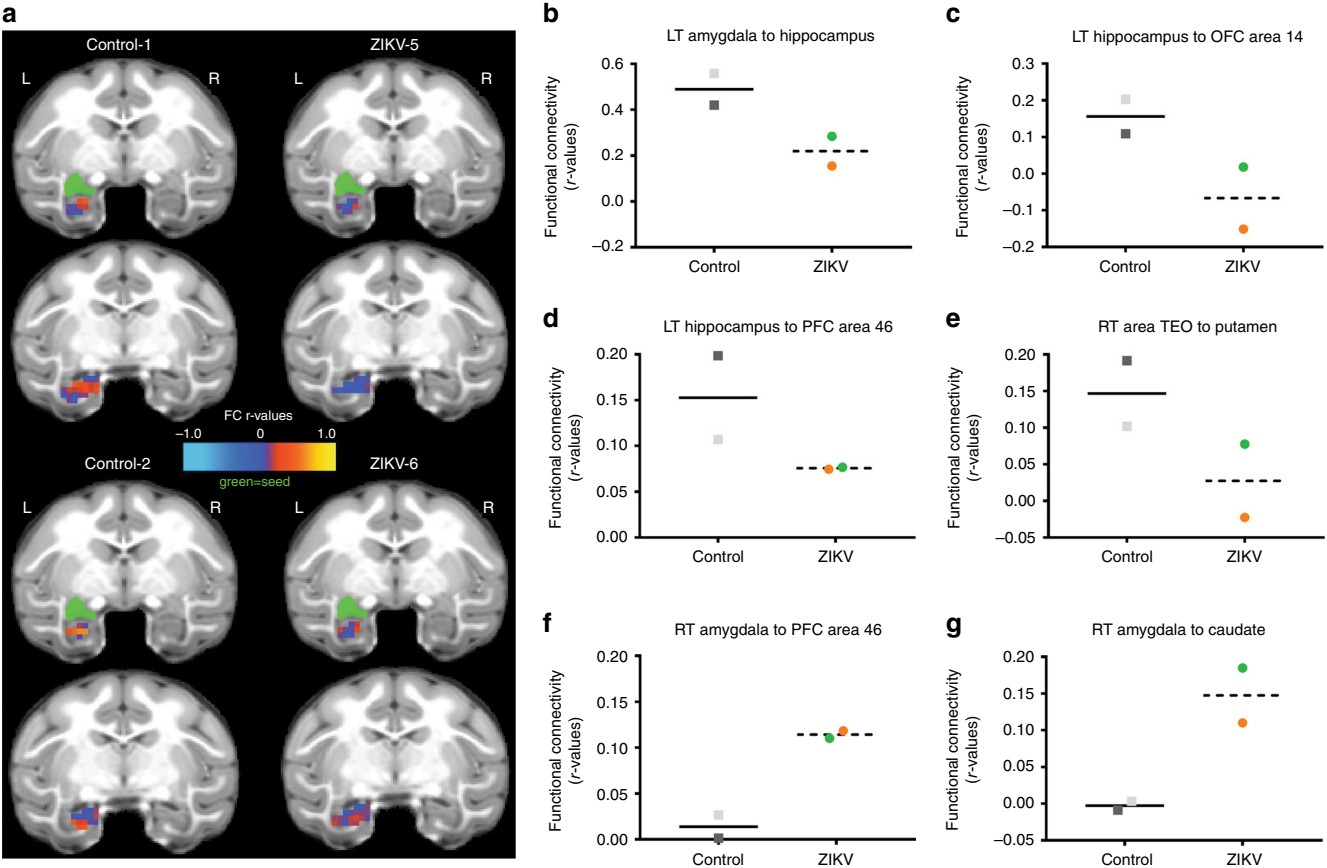

**Fig. 4 Altered functional brain connectivity after postnatal ZIKV infection of infant RMs. a** Depiction of amygdala–hippocampus resting state FC measured by rs-fMRI displayed on the UNC-Emory RM infant atlas[43]. Images show the same two coronal slice series in each subject, with the seed (green) placed in the left amygdala and the FC *r* values in the hippocampus colored according to the scale provided. L left hemisphere, R right hemisphere, *r* values, raw FC correlations. **b–g** Illustrate the rs-fMRI measurements of FC between specific brain regions in ZIKV-infected infant RMs (dashed line with circles; green = ZIKV-5 and orange = ZIKV-6) and controls (solid line with squares; dark gray = Control-1 and light gray = Control-2).

14–15 days after infection, virus was not detected in the CSF[31]. In other experiments, ZIKV has been detected in CSF and brain tissue up to 72 days post-infection and virus- or immune-mediated pathology is apparent post-mortem[31,46,47].

Pathologic evaluation was performed on several regions of brain and spinal cord, including PFC, frontal cortex, parietal cortex, temporal cortex, piriform cortex, occipital cortex, basal ganglia, amygdala, substantia nigra, hippocampus, cerebellum, brainstem and cervical, thoracic, lumbar, and sacral spinal cord and cauda equina. Spinal cord and cauda equina revealed no significant lesions and immunohistochemical analysis of glial, neuronal, and apoptotic markers (GFAP, NeuN, and caspase 3, respectively) in the brain regions above were similar in ZIKV-infected and control RMs. However, ventricle expansion in ZIKV-infected RMs was apparent on neuropathological examination (Fig. 5a, b) confirming the enlarged lateral ventricles detected with MRI. In addition, neuropil and perivascular calcification was detected in the putamen of ZIKV-5 (Fig. 5b, c), which may explain the structural and functional changes detected with in vivo MRI scans. Following doublecortin staining, histological examination of ZIKV-infected RMs revealed under-developed dendritic branching of immature neurons in the paralaminar nucleus of the amygdala (Fig. 6), while no abnormalities were observed in the hippocampus. These results, obtained remote from the time of infection, differ from the increased astrogliosis, perivascular inflammation, and apoptosis found in ZIKV-infected RMs after only 2 weeks of infection[31].

Overall, these findings demonstrate that despite ZIKV being cleared from the periphery quickly, the infection can result in brain lesions that impact the long-term growth and functioning of the brain in the absence of ongoing inflammatory changes.

## Discussion

The current study demonstrates that ZIKV infection during the early postnatal period can have long-lasting impact on brain structure, function, behavior, and cognition in primates. To date, most research has focused on congenital ZIKV syndrome[4–6,48], leaving many questions unanswered about the potential impact of postnatal infection on the developing brain. Given the protracted development of the brain in humans and nonhuman primates, there is considerable opportunity for plasticity and functional compensation after early brain damage. Yet, the current study suggests persistent alterations in brain growth and function as a result of damage from early postnatal ZIKV infection with limited recovery.

While ZIKV infection did not impact the RM infants' ability to produce species typical social and emotional behaviors, the expression of those behaviors was altered as compared to controls. We previously reported that ZIKV-infected infant RMs exhibited altered emotional reactivity on an acute stress test as compared to controls[31]. The current findings show that this increased emotional reactivity persisted through the juvenile period, such that ZIKV-infected RMs took longer to habituate to

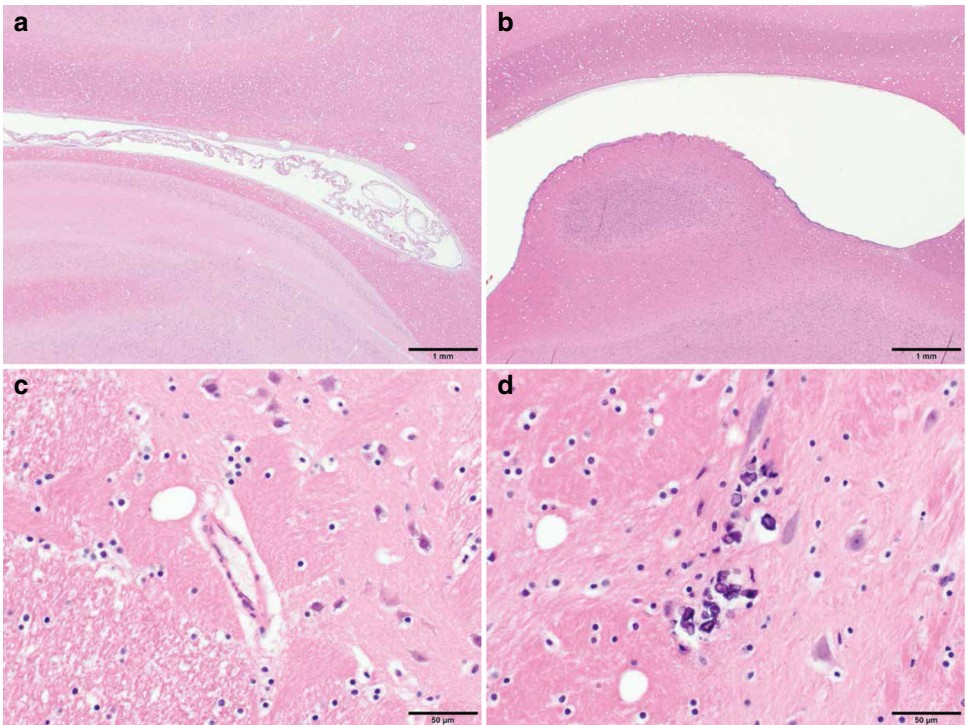

**Fig. 5 Neurohistopathology at 12 months of age after postnatal ZIKV infection of infant RMs. a** Hematoxylin and eosin (H&E) staining of occipital cortex of control animal, ×20 magnification; **b** H&E staining showing distention of lateral ventricle in a occipital cortex of ZIKV-infected RM, ×20 magnification; **c** H&E staining of basal ganglia (putamen) of control animal, ×400 magnification; **d** areas of neuropil calcification (arrowhead), H&E staining, ×400 magnification.

a novel social environment and avoided spending time in the center of the cage as compared to controls. Beyond emotional responses, we were also able to detect alterations in social interactions. Young RMs spend most of their time with other conspecifics engaged in play and affiliative behaviors[49]. Yet, ZIKV-infected RMs preferred to spend more time apart and engaged in fewer affiliative behaviors with a highly familiar partner as compared to controls. Pharmacologically increasing amygdala activity results in decreased social interaction in monkeys[50], which provides a potential mechanism to explain our results showing increased FC between the amygdala and prefrontal area 46, as well as the amygdala and caudate among ZIKV-infected infant RMs. In addition, previous studies of RMs with selective lesions of the amygdala or hippocampus during infancy report similar alterations in socioemotional behavior[51,52], further supporting the notion that behavioral changes in the current study may be the result of structural and functional alterations found in these limbic regions after postnatal ZIKV infection. Overall, these findings indicate that ZIKV infection during this early and vulnerable period of rapid postnatal brain development can have a lasting impact on socioemotional behavior.

Neuroimaging at 12 months of age revealed a persistent ventricular enlargement in ZIKV-infected infant RMs as compared to age-, sex-, and rearing-matched controls. Increased lateral ventricle size was first reported on structural MRIs taken at 3 and 6 months of age[31] and is still present at 12 months of age. Lateral ventricle size was not related to individual variability in brain size, since there was no difference in total brain volume. Importantly, enlarged ventricles for ZIKV-infected infant RMs were also confirmed with neuropathology. These results are similar to in utero ZIKV infection which reports more extensive ventriculomegaly in live born infants and autopsied fetuses[4,5,53,54].

Interestingly, no differences were detected in total WM or GM between ZIKV-infected RMs and controls at 12 months of age,

unlike our previous MRI investigations at 3 and 6 months of age[31]. Previous studies in human and nonhuman primates have identified changes in WM tracts after early life adversity[55,56] and other viral infections[57,58]. Similarly, postnatal ZIKV infection could result in microstructural changes in WM tracts. However, to detect such changes a more sensitive neuroimaging approach, such as diffusion tensor imaging (DTI), would be needed.

Temporal lobe structures are particularly sensitive to early insults, which was also apparent with early postnatal ZIKV infection. Although no differences were detected in amygdala volume at 3 and 6 months of age, by 12 months RM infants exposed to ZIKV had much smaller amygdalae compared to controls. The rapid maturation of the amygdala seen in infant RMs may have precluded the detection of group differences in amygdala volume until after the rate of growth stabilized at around 8 months of age[35,36]. ZIKV-infected RMs also exhibited underdeveloped dendritic branching of immature amygdala neurons, likely limiting their ability to be incorporated into the neural network. These structural changes may explain the differences detected in amygdala FC of ZIKV-infected infants, as well as their increased emotional reactivity and decreased prosocial behavior. The hippocampus has a more protracted development, doubling its volume over the first 2 years of life and continuing to grow into adulthood in RMs[35]. This prolonged neural plasticity in the hippocampus may explain the relative lessening in hippocampal volume differences between RM infants infected with ZIKV and controls at 12 months. ZIKV-infected infant RMs continued to have a smaller hippocampal volume similar to our previous reports at 3 and 6 months of age[31], but the magnitude of the difference between the groups appears to be less pronounced by 12 months of age. While this finding may suggest a potential for recovery or compensation during this period of neural plasticity and rapid brain development, resting state fMRI results demonstrate persistent altered FC between the

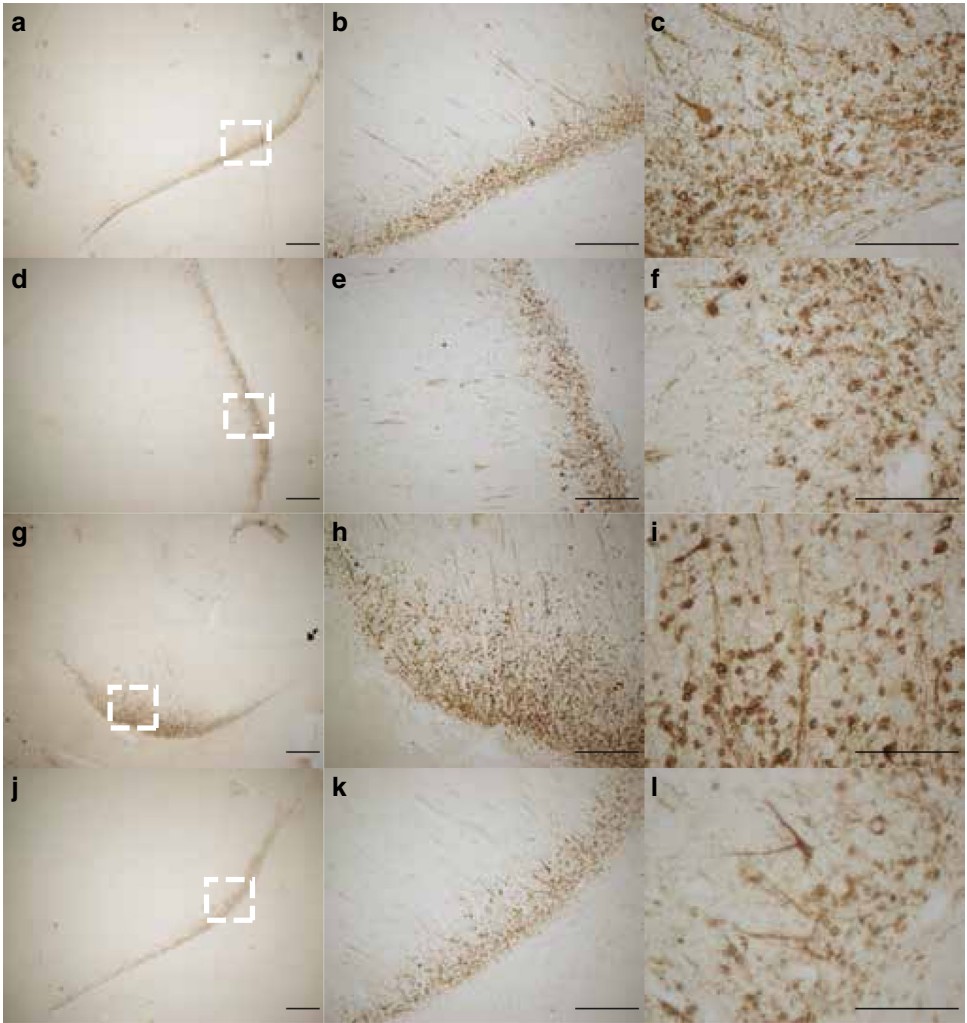

**Fig. 6 Postnatal ZIKV infection decreased dendritic arborization in immature amygdala neurons.** Doublecortin staining in the paralaminar nucleus of the amygdala representative sections shown for ZIKV-5 (**a–c**), ZIKV-6 (**d–f**), Control-1 (**g–i**), and Control-2 (**j–l**). Left panels (**a, d, g, j**) are taken at ×2.5 magnification and the white dotted box is the location of the higher magnification images. Middle panels (**b, e, h, k**) are taken at ×10 magnification. Right panels (**c, f, i, l**) are taken at ×40 magnification. Scale bars 2.5× = 500 μm; 10× = 250 μm; 40× = 100 μm.

hippocampus and other areas important for socioemotional and cognitive functioning, despite some structural normalization.

The current study also found structural and functional differences in the striatum (i.e. putamen and caudate) on MRI, as well as calcification on neuropathology. Calcification in subcortical areas have been reported in several studies of confirmed and suspected congenital Zika syndrome cases in humans[20,39,59]. These findings as well as the smaller cerebellum GM volumes may explain the increased incidents of balance loss seen amongst the ZIKV-infected infant RMs during their exploration of a novel social environment. Although we did not perform a specific test of gross motor function, our current findings of uncoordinated movements resulting in losses of balance suggest that postnatal ZIKV infection may have impacted motor development and psychomotricity, which has also been demonstrated in a murine model of postnatal ZIKV infection[32]. Two factors could have contributed to the increased loss of balance detected in the ZIKV-infected RMs over the 6 days of social observations. First, as the animals habituated to the large novel play cage, they did more jumping and playing which provided more opportunities for missteps and losses of balance when jumping and landing long distances (5–8 feet) in the play cage. Secondly, as shown in

Fig. 1h, i, ZIKV-5 began to exhibit increased hostility while ZIKV-6 exhibited more submissive behaviors across the days of testing. The hostile chasing and resultant fleeing we observed in the ZIKV-infected animals is a quick behavioral sequence that may have also provided more opportunities for missteps and losses of balance among the ZIKV-infected group.

Structural and functional changes in the hippocampus and striatum may also explain the memory deficits detected on the VPC task at 12 months of age, which were not detected at 6 months of age[31]. In fact, infant RMs infected with ZIKV performed poorly only at 120 s delays indicating a delay-dependent deficit, which is similar to a late-developing delay-dependent memory deficit detected in neonatal hippocampal lesioned RMs[60]. Performance on spatial relational VPC task (e.g. Object in Place) is late developing and sensitive to neonatal and adult hippocampal damage[34,61]. The deficits in spatial relational memory observed in both control and ZIKV-infected RMs were age-appropriate. However, the deficit on the Object Control task for ZIKV-infected RMs could be due to the structural and functional changes detected in the hippocampus and striatum, brain areas important for cognition. Despite the potential for neural plasticity and recovery, postnatal ZIKV infection may have

a long-term impact on cognitive function that emerges with age. Considering the protracted development of the brain and memory systems, we would further hypothesize that postnatal ZIKV infection may impact spatial relational memory as well as other later developing memory systems, such as working memory.

There are limitations to the current study. First, the small number of animals studied made most group comparisons descriptive in nature. As a result, this study should be viewed as an exploratory, proof-of-concept experiment to inform future studies on the neurodevelopmental outcomes of ZIKV infection after birth[62]. The strength of this study does not reside in statistical significance but rather in the integration of neuroscience experiments at multiple scales[63] and its relevance to an understudied clinical problem. Second, our study focused on investigating a single strain of ZIKV, PRVABC59, that has been widely used for RM studies and that is closely related to isolates from Brazil[64], but which likely does not capture the entire spectrum of ZIKV infections. Third, only female subjects were studied, and there may be sex differences in the impact of ZIKV infection as has been recently reported in mice[65]. Fourth, there are several limitations to rs-fMRI which should be noted: (1) FC was obtained under anesthesia, but we note that isoflurane doses were kept below those used in previous studies reporting patterns of coherent BOLD fluctuations similar to the awake state, including in sensory and visual systems[66]; and (2) our design does not allow to establish causality between the alterations in structural and functional brain measures and behavioral/cognitive alterations or to specific physiological, cellular, or molecular brain maturational processes, which will need to be addressed in future studies. Additional studies are needed to substantiate our findings regarding postnatal ZIKV infection, which should include larger sample sizes of males and females and multiple virus strains reflecting the diversity of this infection.

In summary, ZIKV infection during the early postnatal period is associated with long-term alterations in socioemotional behavior and cognitive functioning, as well as changes in brain growth and function. The impact of ZIKV on the brain was also apparent with neurohistopathology conducted at approximately 12 months post-infection, further indicating the long-lasting impact of the virus on neural development. The current study demonstrates that behavioral and cognitive abnormalities become more apparent with age; however, future studies are needed to understand the impact of early postnatal ZIKV infection during later stages of life (e.g. adolescence and adulthood). This RM model sheds light on the potential outcomes of human infants infected with ZIKV postnatally and provides a platform to test therapeutic approaches to alleviate the neurologic consequences of ZIKV infection.

## Methods

**Study design**. The objective of this study was to evaluate the long-term neurodevelopmental outcome of postnatal ZIKV infection of infant RMs. After infection, animals underwent virologic and immunologic measurements, as well as neuroimaging, behavioral assessments, and neurohistopathology. This was an exploratory pilot study of postnatal ZIKV infection in infant RMs; therefore, a formal power analysis was not performed. Infants were randomized to either the ZIKV or control groups based on the day of birth. Veterinary and animal resource staff were not blinded to study group due to the necessary environmental precautions that were taken with ZIKV. Laboratory staff were blinded to study group during the conduct of experiments and analyses. Methods have been previously published[31], but will be briefly described below.

**Animals**. Four female infant Indian RMs (*Macaca mulatta*) were used in this study, two were exposed to ZIKV at 5 weeks of age, while the remaining two served as age-, sex-, and rearing-matched uninfected controls. The infants were delivered naturally by their dams while housed in indoor/outdoor social groups. Infants were then removed from their dams at 5–10 days of age and transported to the Yerkes National Primate Research Center (YNPRC) nursery facility. Maternal factors (age, body weight, immune) have been shown to impact infant temperament, behavior,

and immune responses[67–70], thus the dams in this study were similar between the groups. Dams had a range in maternal age (5.5–18 years) and parity (3–12 infants), but this was spread equally between dams of control infants (maternal age: 7 and 13 years; parity: 3 and 8) and dams of ZIKV-infected infants (maternal age: 5.5 and 18 years; parity: 3 and 12). All dams had similar body weights (pre-pregnancy: 6.76–8.55 kg; pregnancy 10.76–12.1 kg), were from the specific pathogen-free colony (negative for Herpes B, SIV, SRV, and STLV1), had not been previously used for infectious diseases or vaccine studies, and did not have any clinical signs of infection during pregnancy. All infants were singly housed in warming incubators for the first 3 to 4 weeks, but had visual and auditory contact with conspecifics. Infants were hand-fed formula every 2–3 h for the first month, then via self-feeders for the next 3 months as per standard YNPRC protocol. Soft blankets, plush toys, or fleece were provided and changed daily. Soft chow and fruits were introduced starting at 1 month of age, and by 4 months, formula was discontinued and all were fed a diet of Purina Primate Chow, Old World Monkey formulation, supplemented with daily fruits and vegetables. Water was provided ad libitum. At 4 weeks of age, infants transitioned into age-appropriate caging with visual and auditory contact with conspecifics. At 6–7 weeks of age, they began socialization consisting of protected contact housing supplemented with 2–3 h daily of full contact with their age-matched peer. Infants were pair housed according to their treatment group (ZIKV or control), as in other infectious diseases studies in infant RMs[71,72]. Specifically, ZIKV-infected RM infants were paired together and likewise for the uninfected controls. By 3 months of age, each pair was housed entirely together. ZIKV-infected infants were housed in an ABSL-2+ (Animal Biosafety Level 2) room until cleared of virus in blood and urine. Control infants were initially housed in a separate ABSL-2 room, and then all infant pairs were housed together after ZIKV clearance with visual contact to increase visual socialization and control for environmental factors. Precautions were taken to ensure that control and ZIKV-infected infants received similar housing, nutrition, and enrichment. In addition, control infants were subjected to the same sedation and collection procedures (blood and urine) as ZIKV-infected infants to minimize any differences in exposure to stressful experiences. All housing was indoors on a 12 h light–dark cycle. This study was conducted in strict accordance with U.S. Department of Agriculture regulations and the recommendations in the Guide for the Care and Use of Laboratory Animals of the National Institutes of Health, approved by the Emory University Institutional Animal Care and Use Committee, and conducted in an AAALAC accredited facility.

**Virus and infection**. The ZIKV of Puerto Rican origin (PRVABC59, GenBank accession number: KU501215.1) used in this study had been passaged four times and titrated on Vero cells. Experimental infections were performed via the subcutaneous route using $10^5$ pfu of ZIKV PRVABC59.

**Behavioral assessment**. To examine social and emotional behavior, subjects were tested in highly familiar dyads similar to previously published protocols[50,73] in a large novel social play cage (390 cubic feet). The social play cage was a hexagonal nonhuman primate social cage (Flex-A-Gon; Britz and Company, Wheatland, WY). There were six separate days of 45 min of testing for a total of 4.5 h of behavioral observations. Subjects entered the cage at the same time, but from separate doors that alternated sides of the cage each day. Each test day was videotaped for behavioral coding later using the Observer XT 14 software (Noldus Inc.) and a detailed ethogram (Supplementary Table 4). One experimenter coded all of the videotapes with a high degree of intra-rater reliability Cohen's $\kappa = 0.98$; and an average inter-rater reliability of Cohen's $\kappa = 0.86$ with other trained experimenters at YNPRC. Since the experimenter was not blind to the animals' treatment, videos were coded without identifying information about the subject, which was revealed after coding was complete.

**Cognitive assessment**. A Tobii T60/T120 eye tracker (Tobii Pro Technology, Reston, VA) was used for VPC testing. Subjects were acclimated to a size-appropriate nonhuman primate chair and to facing the eye tracker monitor on which visual images were displayed. Visual stimuli were displayed on the eye tracker monitor using Tobii Studio™ software (version 3.4.8). The tester running the session was separated visually from the monkey by a curtain. The tester could monitor whether the tracker was detecting the eyes, and a gaze-trail was displayed over a view of the presented stimuli on the tester's laptop. Infants were calibrated to the eye tracker at the beginning of each session.

Existing sets of visual stimuli for infant RMs were used comprised of photo/clip-art images of everyday junk objects (Nova Art Explosion 800,000 ClipArt, Nova Development, Calabasas, CA). Selected images for each set were chosen to be as different as possible. As shown in Fig. 2a, each testing trial consisted of a Familiarization phase, in which a sample image was presented in the center of the screen. The image remained on screen until the monkey had cumulatively fixated within the boundaries of the image for a total of 15 s (as timed by the tester using a stopwatch). At this point, the image disappeared and the screen remained black for the delay time (10, 30, 60, or 120 s). The delay period was followed by the first of two retention tests, during which the sample image and a novel image were placed on either side of the screen for 5 s and the RM was free to look at either. Both images then disappeared for another 5 s, followed by the second retention test in

which the left–right positions of the sample and novel images were switched to prevent against side bias. Initial placement of the objects was determined pseudorandomly such that the sample object appeared equally often on either side of the screen for the first retention test. Each trial was separated by a 30 s inter-trial interval. Subjects completed a total of 40 trials, 10 per delay.

Total duration of the fixation times to each predetermined region of interest (ROI) was calculated using Tobii Studio™ software. Each ROI encompassed the image and the white background and was categorized as sample, for familiarization stimuli, and novel or familiar for retention stimuli. The total fixation duration time within each ROI was compared, yielding a total time looking at novel and/or familiar stimuli. In this task, RMs (like humans) have an innate tendency to look at the novel stimulus, and so Preference for Novelty is a standard measure of memory. Preference for Novelty score was calculated as [looking time (novel)/total looking time (novel + familiar) × 100]. Trials were included in the analysis if fixations were observed during both retentions and if no evidence of side bias was observed in that trial. Trials were averaged per subject to yield a single score for each delay. On average, each subject contributed five usable (range of 2–7) trials at each delay.

**Neuroimaging assessments**. Neuroimaging data were collected 12 months of age using a Siemens 3T Tim Trio system (Siemens Medical Solutions) with a Tx/Rx 8-channel volume coil. Data were acquired in a single session, which included T1- and T2-weighted structural MRI scans, and two 15-min echo-planar imaging (EPI) rs-fMRI (T2*-weighted) scans to measure temporal changes in regional BOLD signal. Animals were scanned in the supine position in the same orientation, achieved by placement and immobilization of the head in a custom-made head holder via ear bars and a mouthpiece. After initial telazol induction and intubation, scans were collected under isoflurane anesthesia (0.8–1%, inhalation, to effect), kept at the lowest dose possible to minimize effects of anesthesia on BOLD signal[66]. End-tidal CO2, inhaled CO2, O2 saturation, heart rate, respiratory rate, blood pressure, and body temperature were monitored continuously and maintained during each MRI session.

High-resolution T1-weighted MRI scans were acquired for volumetrics and registration of the functional scans using a three-dimensional magnetization-prepared rapid gradient echo (3D-MPRAGE) parallel image sequence [repetition time/echo time (TR/TE) = 2600/3.46 ms; field of view (FOV), 116 mm × 116 mm; voxel size, 0.5 mm$^3$ isotropic; eight averages] with GeneRalized Autocalibrating Partially Parallel Acquisitions (GRAPPA) acceleration factor of $R = 2$. T2-weighted MRI scans were collected using a 3D fast spin-echo sequence (TR/TE = 3200/373 ms; FOV, 128 mm × 128 mm; voxel size, 0.5 mm$^3$ isotropic; three averages) with GRAPPA ($R = 2$) to aid with registration and delineation of anatomical borders. Structural MRI data processing and analysis data sets were processed using AutoSeg_3.3.2 segmentation package[74] to get the volumes of brain WM and GM, CSF, and cortical (temporal visual area, temporal auditory area, and prefrontal, frontal, parietal, occipital lobes, and cerebellum) and subcortical (hippocampus, amygdala, caudate, and putamen) brain areas. Image processing steps included the following: (i) averaging T1 and T2 images to improve signal-to-noise ratio, (ii) intensity inhomogeneity correction, (iii) rigid-body registration of the subject MRI to the 12-month UNC-Emory infant RM atlases[43], (iv) tissue segmentation and skull stripping, (v) registration of the atlas to the subject's brain to generate cortical parcellations (affine followed by deformable ANTS registration), and (vi) manual editing of the amygdala and hippocampus done on all scans using previously published neuroanatomical boundaries[35]. For more detailed description of the structural MRI analysis, see ref. [74]. Cortical and subcortical volume measurements (in cubic millimeters) were corrected by TBV and then multiplied by 100 (for example, hippocampal volume/TBV × 100).

Resting state functional MRI was also conducted. BOLD-weighted functional images were collected using a single-shot EPI sequence (2 × 400 volumes; TR/TE = 2060/25 ms, 2 × 15 min; voxel size, 1.5 mm$^3$ isotropic) after the T1-MRI scan. An additional short, reverse-phase encoding scan was also acquired for unwarping susceptibility-induced distortions in the EPI images using previously validated methods[75].

Functional MRI data were preprocessed with an in-house pipeline in Nipype[76] with modifications[77], including adaptations for the RM brain[78]. After the EPI functional time series were concatenated and rigid-body registered to the subject's T1-weighted structural image, this was transformed to conform to age-specific T1-weighted RM infant atlases[43] using nonlinear registration methods in FSL (FNIRT). These atlases were registered to the 112RM-SL atlas in F99 space[79]. We registered the scans to the age-specific appropriate infant atlas (12 months). Frames with displacement values greater than 0.2 mm were removed[80], and data were visually inspected upon preprocessing completion to exclude series with unsatisfactory co-registration or significant BOLD signal dropout.

FC between ROIs, including the amygdala, hippocampus, PFC (dorsal, medial, and ventral areas), OFC, and TEO, were analyzed. ROIs were defined on the basis of published anatomical parcellations[42] mapped onto the UNC-Emory RM infant atlases registered to the F99 space[43]. The left and right amygdala automatic label maps were manually adjusted by experts using cytoarchitectonic maps in the existing UNC-Wisconsin adolescent atlas[81], and other ROIs were manually edited in the infant atlases following established anatomical landmarks[82]. The time course of the BOLD signal was averaged across all voxels within each ROI using a single ROI as the seed and then correlated with the time course of the other ROIs. FC values are calculated as correlation coefficients ($r$ values) between ROI BOLD time courses, which were extracted from correlation matrices using MATLAB version R2019a (MathWorks Inc., Natick, MA).

**Histopathological assessments**. At the study end point, the animals were euthanized and a complete necropsy was performed. For histopathologic examination, brain and spinal tissue samples were fixed in 10% neutral-buffered formalin, routinely processed, paraffin-embedded, sectioned at 4 μm, and stained with hematoxylin and eosin. To define specific cell types, we performed immunohistochemical (IHC) analysis by using antibodies against Glial fibrillary acidic protein (Agilent Technologies, M0761), NeuN (EMD Millipore, MAB377), caspase 3 (Cell Signaling, 9662), and doublecortin (Santa Cruz, SC-8066). IHC staining on sections of brain from both ZIKV infected and control animals was performed using a biotin-free polymer system (Biocare Medical). The paraffin-embedded sections were subjected to deparaffinization in xylene, rehydration in graded series of ethanol, and rinsed with double distilled water. The brain sections were incubated with rabbit anti-human glial fibrillary acidic protein and caspase 3 and mouse anti-human NeuN antibodies for 1 h after heat-induced epitope retrieval. Antibodies labeling was visualized by development of the chromogen (Warp Red or DAB Chromogen Kits; Biocare Medical). Amygdala and hippocampal sections were pretreated (3% hydrogen peroxidase and 20% methanol in PBS) for 1 h and incubated overnight in anti-goat doublecortin (1:400 dilution; Santa Cruz #sc-8066) at room temperature. Sections were then incubated in horse anti-goat secondary antibody (1:200 dilution; Vector Labs #BA-9500) for 1 h, followed by an hour in ABC (Vector, PK6100) and visualized with DAB (Sigma). Digital images of brain were captured at ×20 and ×400 magnification with an Olympus BX43 microscope equipped with a digital camera (DP26, Olympus, Center Valley, PA) and evaluated using Cellsens® digital imaging software 1.15 (Olympus). A grid method was used to assess fiber density in sections. All sections were independently reviewed by two experts in nonhuman primate brain pathology.

**Reporting summary**. Further information on research design is available in the Nature Research Reporting Summary linked to this article.

## Data availability
All data used for this paper are available from the authors on request. Supplemental materials contains data tables for social behavior (Supplementary Table 1), structural MRI (Supplementary Table 2), and rs-fMRI (Supplementary Table 3).

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

## Acknowledgements

The authors would like to thank Stephanie Ehnert and the YNPRC Division of Research Resources for expert assistance with animal procedures and Shanice Wilson, B.S. for her assistance with social behavior assessments. Funding for this study was provided by the Pilot Grant Program of the Yerkes National Primate Research Center, which is supported by the National Institutes of Health's Office of the Director, Office of Research Infra-structures Programs, P51 OD011132, and the Center for Childhood Infections and Vaccines of Emory University and Children's Healthcare of Atlanta (to A.C.).

## Author contributions

A.C. designed the study with input from M.M., J.R., M.C.A., and M.S.S. M.S.S. provided the virus. J.R. and M.C.A. performed the social and cognitive assessments and interpreted the data. J.R., Z.K.-B., and M.M.S. interpreted the neuroimaging. D.F., E.F., E.E., and M.S. provided tools for imaging analysis. S.G. and M.W.B. performed histopathology. C.M., J.H., C.M., and M.M. processed samples to confirm viral infection. S.M.J. and J.K.C. provided animal care and collected samples. A.C., J.R., and M.M. wrote the manuscript with help from the other authors.

## Competing interests

The authors declare no competing interests.
