## [Peer Review File · Nature Communications]

Reviewers' comments:

Reviewer #1 (Remarks to the Author):

Thank you for the opportunity to review this exciting work. The manuscript was revised well per my previous comments. This is a well-written manuscript and engaging to read. The results are fascinating and highlight the critical need to fully understand the risk for adverse child neurodevelopment from both prenatal and also postnatal exposure to ZIKV. This project thoughtfully used a multi-modal approach involving infant neurobehavioral and cognitive assessment, structural and functional brain MRI, and neuropathology to describe long-term neurologic changes in infant macaques exposed to ZIKV at 5 weeks of postnatal age. Since 1 year old infant macaques are developmentally similar to a 4-5 year old human child, this study can help to inform neurobehavioral abnormalities that we can only now start to assess in human children exposed to the virus during the epidemic years of 2015-2016. Since the majority of human cohorts have focused on prenatal exposure, the finding that postnatal exposure can have these types of consequences is very important to public health and warrants careful study in human postnatally exposed children and long term study.

Sarah B. Mulkey, MD, PhD

Reviewer #4 (Remarks to the Author):

Long-term alterations in brain and behavior after postnatal Zika virus infections in infant 2 macaques

The study reports the behavioral and neuropathological effects of neonatal ZIKV infection in two infant rhesus macaques. The study is well-done and comprehensive and the findings are compelling implicating ZIKV infection on behavioral developmental alterations in young rhesus infants. The main limitations of the study are the small sample size. Aside from the behavioral, motor, and memory deficits reported—the structural alterations in the brains of the two ZIKV infected infants is quite compelling as are the neuropathologic changes (dystrophic calcifications) reported in one ZIKV infected infant. The studies findings are important although it is unclear whether the changes reported would have resulted in long-term alterations in development given the plasticity of the nervous system at this young age. Despite the authors efforts to control for possible differences in ZIKV infected and control infants, the extremely small n makes any changes (although concordant) highly suspect as chance findings especially if the ZIKV infants were co-housed as a conspecific pair.

Minor comments:

- It is unclear if the age-matched animal was ZIKV infected—were the two ZIKV infected macaques paired together? It seems given the small n, that this could be a major confounder for the behavioral studies.
- Likewise, what were the characteristics of the dams of the ZIKV infected infants as compared to the controls? Maternal age and other factors may have contributed to gestational differences that could serve as confounders.
- Figure 6: differences in dendritic cell branching are based on what? Scoring, image analysis—data is not shown.

Point-by-point response:

Point 1: *It is unclear if the age-matched animal was ZIKV-infected – were the two ZIKV infected macaques paired together? It seems given the small n, that this would be a major confound for the behavioral studies.*

Response: The two ZIKV-infected infants were pair housed after their initial viremia had cleared from plasma and their urine was also documented to be negative for ZIKV. Likewise, the two age-, sex-, and rearing-matched control infants were pair housed together starting at the same age. Both pairs of infants were therefore highly familiar social partners during testing, which is commonly used paradigm in behavioral neuroscience studies (*Emery et al., Behav Neurosci 2001; Bauman et al., J Cogn Neurosci 2004; Malkova et al., Behav Neurosci 2010; Moadab et al., Behav Neurosci 2015; Wellman et al., J Neurosci 2016*). The differences in prosocial behavior between ZIKV-infected and control infants is even more striking considering that the tests were conducted in highly familiar social partner pairs. A mixed group (ZIKV and controls) social housing design was not chosen for this study to avoid the potential confounder of cross contamination (ie, ZIKV infection) of the control infants as at the time this study was conducted it was unknown if ZIKV persisted in infant macaques. However, once ZIKV-infected infants had cleared the virus from plasma within the first 2 weeks after infection, both pairs of infants were housed in the same room for the following 11 months allowing for visual and auditory access to one another. Therefore, the ZIKV-infected and control infants were not housed in social isolation to other animals and were matched for experience throughout development (*Mavigner et al., Sci Trans Med 2018*). Importantly, pair housing animals according to their treatment is common in infectious diseases studies in infants (*Mavigner et al., J Virol 2018; Carryl et al., Brain Sci 2017; Jensen et al., J Virol 2016*), whereas adult macaques are typically singly housed for the duration of these experiments.

Point 2: *What were the characteristics of the dams of the ZIKV infected infants as compared to the controls? Maternal age and other factors may have contributed to gestational differences that could serve as cofounders.*

Response: Reviewer #4 brings up an interesting point that maternal factors have been linked to differences in socioemotional behavior and temperament in humans and animal models. In fact, high cortisol levels in maternal milk are associated with a nervous temperament in daughters (*Hinde et al., Behav Ecol 2015*), but more confident temperament in sons (*Sullivan et al., Dev Psychobiol 2011*). We also are aware that cortisol levels in milk decline with maternal age/parity in RMs. In the current study all infants were breastfed for 7-10 days before being transferred to nursery-reared housing for the remainder of the experiment. The dams in our study had a range in maternal age (5.5-18 years) and parity (3-12 infants), but this was spread equally between dams of control infants (maternal age: 7 & 13 years; parity: 3 & 8) and dams of ZIKV-infected infants (maternal age: 5.5 & 18 years; parity: 3 & 12). Maternal obesity/increased BMI have been associated with inattention, ADHD, and emotional dysregulation (*Chen et al., Int J Epidemiol 2014; Kong et al., Pediatrics 2018*). However, the RMs dams in our study had similar pre-pregnancy (6.76-8.55 kg) and pregnancy (10.76-12.10 kg) body weights. Lastly, maternal

immune responses have been linked to changes in social behavior (*Bauman et al., Biol Psychi 2014*). However, all dams used in this study were from the SPF colony (negative for Herpes B virus, SIV, SRV, and STLV1), had not been previously used for infectious diseases or vaccine studies, and did not have any clinical signs of infection during pregnancy so major differences are not anticipated between the dams for the control and ZIKV-infected groups. Measuring maternal immune responses is beyond the scope of the current study.

Point 3: Figure 6: differences in dendritic cell branching are based on what? Scoring, image analysis—data is not shown.

Response: We thank Reviewer #4 for giving us the chance to clarify our methods. All pathologic specimens were independently reviewed by a pathologist at the Yerkes Primate Center as well as an expert in neuroanatomical development process and plasticity in primates. These individuals used a grid method to assess fiber density in sections following doublecortin staining to identify immature neurons. The analysis of both experts revealed a virtual absence of dendrites and processes (eg, dendritic arborization) in the ZIKV-exposed macaques as compared to controls in the paralamina nucleus of the amygdala. These images are shown in Fig. 6.